# Sensitivity Testing of Natural Antifungal Agents on *Fusarium fujikuroi* to Investigate the Potential for Sustainable Control of Kiwifruit Leaf Spot Disease

**DOI:** 10.3390/jof8030239

**Published:** 2022-02-27

**Authors:** Tingting Chen, Xia Wu, Yunyun Dai, Xianhui Yin, Zhibo Zhao, Zhuzhu Zhang, Wenzhi Li, Linan He, Youhua Long

**Affiliations:** 1Research Center for Engineering Technology of Kiwifruit, Institute of Crop Protection, College of Agriculture, Guizhou University, Guiyang 550025, China; gzctt126@126.com (T.C.); xwu2500109774@163.com (X.W.); dyy2152096850@163.com (Y.D.); xhyin@gzu.edu.cn (X.Y.); zhaozhibozhi@hotmail.com (Z.Z.); zhuzhuzhang9612@126.com (Z.Z.); lwz9512@126.com (W.L.); gzhln9618@126.com (L.H.); 2Teaching Experimental Field of Guizhou University, Guizhou University, Guiyang 550025, China

**Keywords:** kiwifruit leaf spot, *Fusarium fujikuroi*, natural antifungal agents

## Abstract

Kiwifruit is a nutritious and economically important fruit that is widely cultivated in China. In 2021, leaf spot disease of kiwifruit was discovered in the main kiwifruit-producing area of Xifeng County, Guizhou Province, China. Leaf spot disease weakens plant photosynthesis and reduces nutrient synthesis, thereby affecting plant growth. We studied the morphological characteristics and performed a combined analysis of *EF-1**α*, *RPB2*, and *TUB2* genes of *Fusarium fujikuroi,* a fungus associated with leaf spot disease. The pathogenicity of *F. fujikuroi* followed Koch’s hypothesis, confirming that this fungus is the cause of kiwifruit leaf spot disease. The sensitivity of seven natural antifungal agents against *F. fujikuroi* was measured using the mycelial growth rate method. Honokiol, cinnamaldehyde, and osthol showed good antifungal effects against *F. fujikuroi*, with EC_50_ values of 18.50, 64.60, and 64.86 μg/mL, respectively. The regression coefficient of cinnamaldehyde was the largest at 2.23, while that of honokiol was the smallest at 0.408. *Fusarium fujikuroi* was the most sensitive to cinnamaldehyde.

## 1. Introduction

Kiwifruit (*Actinidia,* Actinidiaceae) is a deciduous and dioecious liana fruit tree that produces economically valuable and highly nutritional fruit. Kiwifruit are popular with consumers as they are rich in vitamins [1], phenolics [2], carotenoids and minerals [3], and antioxidant active ingredients [4,5]. Kiwifruit cultivation is mainly distributed in the US, Chile, New Zealand, Australia, China, and some European countries, and the total production of kiwifruit worldwide is estimated to be approximately 1600 tons per year [6], making it one of the most important commercial fruit crops in the world. After decades of rapid growth of the kiwifruit cultivation industry, China has become the world’s largest producer of kiwifruit [7]. However, many diseases have appeared in kiwifruit plants, and the types of pathogens have become increasingly diverse, including *Botryosphaeria dothidea* [8], *Alternaria alternata* [9], and *Phomopsis longicolla* [10] that cause soft rot; *Pseudomonas syringae* pv. *actinidiae* [11] that causes canker; *Colletotrichum acutatum* [12] that causes anthracnose; and *Pseudomonas viridiflava* [13,14] that causes bacterial blossom blight. These diseases limit the sustainable development of the kiwifruit industry.

*Fusarium fujikuroi* has a wide geographic distribution and host range, and it is one of the most difficult agricultural pathogens to control [15]. It is a hemibiotrophic pathogen [16], belonging to the *Gibberella fujikuroi* species complex of pathogenic fungi, which can produce gibberellins and synthesise fumonisins and fusarin metabolites [17,18,19,20,21]. *Fusarium fujikuroi* is responsible for bakanae disease in rice [22], leaf spot in *Lasia spinosa* [23], and maize ear rot in corn [24], among others. The fungus was first reported to cause diseases in kiwifruit leaves in 2020 [25], and in May 2021, a kiwifruit with leaf spot disease was found in a kiwifruit plantation in Xifeng County, Guiyang City, Guizhou Province, China, with an incidence rate of approximately 23%. The disease seriously inhibited photosynthesis, resulting in declines in both the yield and quality of the kiwifruit. The typical symptoms, including brown necrotic spots, shrinking and curling of leaf margins, and irregular lesions, are observed on the leaves. Identifying the cause of kiwifruit leaf spot disease and its control methods are critical to orchard management. In this study, we aim to understand the pathogenic factors of kiwifruit leaf spot disease and investigate potential natural fungicidal agents, considering the wider applications of these agents for agriculturally significant pathogens.

## 2. Materials and Methods

### 2.1. Diseased Leaf Collection and Isolation Procedures and Natural Antifungal Agents

In May 2021 (kiwifruit flowering period determined using the BBCH-scale), leaves of *Actinidia deliciosa* ‘Guichang’ visibly affected by leaf spot disease were collected from Xifeng County (27°2′ N, 106°30′ E). The leaves were disinfected with 75% alcohol for 30 s, rinsed with sterile water three times, and dried with sterile paper. The diseased tissue was then cut into 1 cm^2^ pieces, placed in potato dextrose agar (PDA: 200.0 g potato, 20.0 g glucose, 17.0 g agar/L) plates, and cultured at a constant temperature of 28 °C and 75% relative humidity for 5 d. A single colony was selected and transferred to new PDA plates until the mycelia were identical in colour and shape. The pure culture obtained was stored in 25% (volume/volume) glycerol at −80 °C for long-term storage. The natural antifungal agents (osthole, cinnamaldehyde, resveratrol, allicin, honokiol, citral, and carvacrol) used with purities of ≥98% were provided by Shanghai Macklin Biochemical Co., Ltd. (Shanghai, China) and stored at 4 °C.

### 2.2. Morphological and Molecular Characterisation

The colony traits of the isolates were evaluated after observing the cultures in PDA plates and synthetic low-nutrient agar (SNA) medium (1.0 g KH_2_PO_4_, 1.0 g KNO_3_, 0.5 g MgSO_4_·7H_2_O, 0.5 g KCl, 0.2 g glucose, 0.2 g sucrose, 0.2 g agar/L) after incubation at 28 °C for 5 d in the dark. The observations of hyphae and conidia were performed and recorded using a binocular microscope (Leica DM500, Leica Microsystems (Shanghai) Trading Co., Ltd., Shanghai, China) equipped with a digital camera, and the isolated strains were subjected to targeted DNA sequence amplification and sequencing. The fungal mycelia were collected and tested using a fungal DNA extraction kit (Tiangen Biotech (Beijing) Co., Ltd., Beijing, China) according to the manufacturer’s instructions, with combined use of the translation elongation factor 1 alpha gene (*EF-1ɑ*), RNA polymerase II second largest subunit (*RPB2*) gene, and beta-tubulin (*TUB2*) gene (Table 1). The polymerase chain reaction (PCR) system consisted of 1 μL each of the forward and reverse primers (10 μmol), 12.5 μL of 2× TaqMasterMix, 2 μL of 50 ng/μL DNA, and ddH_2_O to make up the volume to 25 μL. The relevant PCR procedure was used to amplify the strain, following the methods of O’Donnell et al. [26], O’Donnell et al. [27], and Glass and Donaldson [28]. The captured PCR product image was obtained with (1×) TAE prepared on a 1.5% agarose gel in the BIO-RAD Gel Doc XR + Gel imaging system (BIO-RAD, Hercules, CA, USA) and was sent to Sangon Biotech (Shanghai) Co., Ltd. (Shanghai, China) for sequencing. ContigExpress Application (copyright: 1999–2000 InforMax, a component of Vector NTI Suite 6.0) was used to splice the tested gene sequences, which were uploaded to the GenBank database in NCBI (http://www.ncbi.nlm.nih.gov (accessed on 9 December 2021)) for comparison and analysis. A phylogenetic tree was constructed with the maximum likelihood (ML), maximum parsimony (MP), and Bayesian inference (BI) methods as implemented in the CIPRES Science Gateway (phylo.org) [29] and visually analysed using Fig tree software [30].

### 2.3. Pathogenicity Assays

Twenty-five fungal strains with different colony morphologies were isolated and used for Koch’s hypothesis [25]. All isolated strains were placed in potato dextrose broth (PDB; 200.0 g potatoes, 20.0 g glucose, 1 L water), shaken at 130 rpm and 28 °C for 5 d, and filtered through gauze to collect the conidia. Then, 500 μL of 1 × 10^6^ conidia/mL was collected using a hemacytometer and sprayed evenly on sterilised healthy Guichang kiwifruit leaves with petioles. For a blank control, 500 μL of sterilised distilled water was sprayed. Each petiole was wrapped in wet cotton to prevent the leaves from drying out. The inoculated leaves were placed in a light incubator at 28 °C and 75% relative humidity with a 16/8 h light/dark photoperiod, and the disease progression of the leaves was regularly observed. Three replicates were used for each isolate. Scanning electron microscopy (SEM; SU8010, Hitachi, Tokyo, Japan) was used to observe the growth of conidia on the leaves of kiwifruit; the fungi were separated using the direct separation method, and the isolated strains were identified. The pathogenicity test was performed three times.

### 2.4. Antimicrobial Activity of Natural Antifungal Agents on Mycelial Growth

According to the mycelial growth rate method described by Xin [31], different antifungal natural extracts were dissolved in organic solvents (citral was dissolved in ethanol; honokiol in dimethyl sulfoxide; and osthole, carvacrol, cinnamaldehyde, resveratrol, and allicin in acetone), and then using water, were mixed evenly with the PDA medium at different concentrations. The *F. fujikuroi* colony (with a diameter of 6 mm) was placed in the centre of the PDA medium containing natural antifungal agents and cultured at 28 °C and 75% relative humidity for 5 d under dark conditions, then the colony diameter (mm) was measured using a ruler. The EC_50_ (concentration for 50% of maximal effect) values of different plant extracts were calculated using IBM SPSS analytics (SPSS Inc., Chicago, IL, USA) [32].

## 3. Results

### 3.1. Isolation and Identification of Strain XFT3-1 from Kiwifruit Leaves

Strain XFT3-1 formed a round colony on the PDA medium after 5 d and produced floccose white aerial mycelia with light purple or reddish-pink coloration (Figure 1A,B). Microconidia and macroconidia were observed on the SNA medium after 5 d (Figure 1C–G). The microconidia occurred in the form of long chains. The individual spores were oval-shaped, measuring approximately 6.52–9.83 × 2.18–5.64 μm, and occurred with or without a diaphragm. Few macroconidia were observed. These were spindle- or sickle-shaped and measured approximately 23.39–47.92 × 2.08–5.75 μm with 3–5 separations. The conidiophore was singular and bottle shaped. The morphological identification of strain XFT3-1 as *F. fujikuroi* was consistent with the results of Laurence [33] and Ibrahim [34].

Sequence alignment analysis of strain XFT3-1 based on RNA polymerase II second largest subunit (*RPB2*), translation elongation factor 1 alpha (*EF-1ɑ*), and beta-tubulin (*TUB2*) genes was performed, and a reference strain was obtained from GenBank (Table 2) for construction of the *RPB2*-*EF-1ɑ*-*TUB2* phylogenetic tree [35]. The results showed that, using *Fusarium acutatum* NRRL 13308 as the outgroup, the strain XFT3-1 (GenBank accessions: EF-1ɑ, OL774567; RPB2, OL774568; TUB2, OL774569) and *F. fujikuroi* sampled from various locations were clustered into a single group with a self-sustaining rate of 100% ML, 99.45% MP, and 1.00 BI (Figure 2), which supported the morphological and molecular identification.

### 3.2. Pathogenicity

Leaf spot disease was found in the orchard during the flowering period of kiwifruit, as evidenced by brown necrotic spots on leaves, shrinkage and curling of leaf margins, and irregular lesions (Figure 3A,B). Pathogenicity results showed that 5 d after treatment with the strain XFT3-1 conidia suspension, brown irregular spots appeared on the kiwifruit leaves and the leaf margins withered and shrunk (Figure 3C,D). Therefore, the symptoms after artificial inoculation were similar to those observed in natural plantations. The leaves that received sterile distilled water did not show symptoms of leaf spot disease. A comparison of the microstructure of inoculated leaves revealed that while leaves in the control treatment were clear, smooth, and free of impurities (Figure 3E), leaves inoculated with the conidia of *F. fujikuroi* had a wrinkled and cracked epidermis and the conidia had germinated in the leaf tissue (Figure 3F). To satisfy Koch’s hypothesis, the fungal strains were re-isolated from the infected leaves, and the *TEF1-α*, *TUB2*, and *RPB2* genes from the *F. fujikuroi* XFT3-1 used in the artificial inoculation were found to be grouped together.

### 3.3. Fungicide Sensitivity of Strain XFT3-1

The screening of natural extracts against strain XFT3-1 is conducive to the development of green, low-toxic fungicides that can effectively control *F. fujikuroi.* The results of sensitivity testing of seven natural antifungal agents on *F. fujikuroi* are shown in Table 3. Honokiol had significant antibacterial activity, with an EC_50_ value of 18.50 ± 0.20 mg/L, followed by osthole and cinnamaldehyde, with EC_50_ values of 64.86 ± 0.18 and 64.60 ± 0.23 mg/L, respectively. The antibacterial activity of citral was poor, with an EC_50_ value of 509.25 ± 0.50 mg/L. According to the regression equation, the slope of cinnamaldehyde was the largest at 2.23, and that of honokiol was the smallest at 0.408. This indicated that *F. fujikuroi* was the most sensitive to cinnamaldehyde and the least sensitive to honokiol.

## 4. Discussion

Kiwifruit is a large, dioecious, deciduous woody vine. It is one of four tree species that were successfully domesticated and cultivated as artificial fruit trees in the 20th century [36]. With the global expansion of land under cultivation for agriculture, pathogenic fungi that cause brown spot disease in kiwifruit have been encountered more frequently. *Fusarium tricinctum* [37], *A. alternata* [38], *Pseudocercospora actinidiae* [30,39], *Didymella bellidis* [40], and *Nigrospora sphaerica* [41] can cause brown spots in kiwifruit. In this study, conventional tissue isolation methods and pathogenicity determination tests were used. In total, 25 strains were isolated but only *F. fujikuroi* strain XFT3-1 caused leaf necrosis and irregular lesions, which is consistent with the results of Li et al. [25]. *Fusarium*
*fujikuroi* is also an important pathogenic factor of bakanae disease in rice, fusariosis of pineapple [34], and stem rot of red-fleshed dragon fruit [42]. However, in vanilla, *F. fujikuroi* is non-pathogenic and is an endophyte coloniser or saprophytic fungus [43]. In response to human activity and weather changes, fungi are likely to experience host jump in plants [44]. At present, there are few reports of *F. fujikuroi* on kiwifruit, but its impacts cannot be ignored and measures for timely detection and prevention should be taken.

Chemical control is the main measure for the prevention and treatment of diseases caused by *F. fujikuroi*. Chemical control agents such as succinate dehydrogenase inhibitor and benzimidazole-, triazole-, and imidazole-based fungicides [45,46,47] are used to inhibit mycelial growth and conidia formation and control crop diseases caused by *F. fujikuroi*. Despite chemical fungicides being able to effectively control the occurrence of diseases to a certain extent, if not applied scientifically, their use can cause *F. fujikuroi* to become resistant [48,49]. On the contrary, biological control agents have attracted increasing attention owing to their potential to increase the sustainability of agricultural operations while being compliant with the European IPM framework and organic farming legislation [50]. Thymol (20%) and eugenol (5%) significantly reduced bakanae incidence [51], and *Pleurotus ostreatus* ‘Heuktari’ extract is used as an agriculture seed disinfectant against *F. fujikuroi* [52]. This study tested a range of natural antifungal agents for antifungal activity and found that honokiol, osthole, and cinnamaldehyde were effective in inhibiting the growth of *F. fujikuroi*. However, their success in preventing and controlling kiwifruit leaf spot disease still needs to be verified using field trials. The mechanisms of the antifungal effects of these natural antifungal agents are not yet clear and require further study.

## 5. Conclusions

In this study, the strain XFT3-1 caused leaf spot disease of kiwifruit, which was identified as *F. fujikuroi* by morphological characteristics, molecular biology and pathogenicity verification. At the same time, Screening of 7 natural antifungal agents, honokiol, osthole and cinnamaldehyde have the potential to inhibit the growth of mycelium, and cinnamaldehyde is the most sensitive against *F. fujikuroi* in vitro.

## Figures and Tables

**Figure 1 jof-08-00239-f001:**
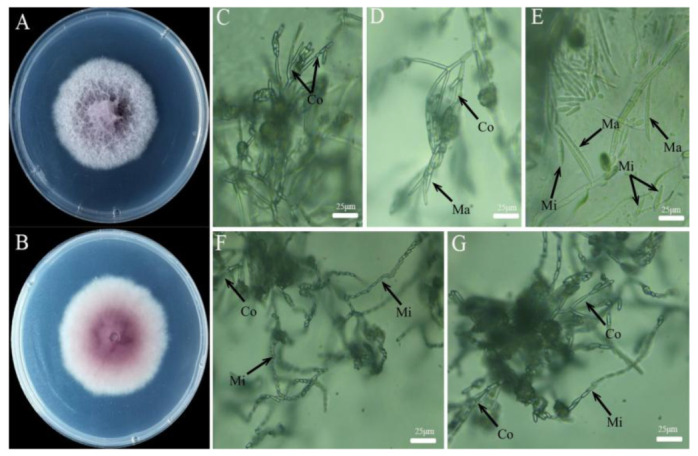
Observation of the colony morphology of strain XFT3-1 on PDA (**A**,**B**) and the conidiophore, macroconidia, and microconidia on SNA (**C**–**G**). Note: Ma: macroconidia, Mi: microconidia, Co: Conidiophore.

**Figure 2 jof-08-00239-f002:**
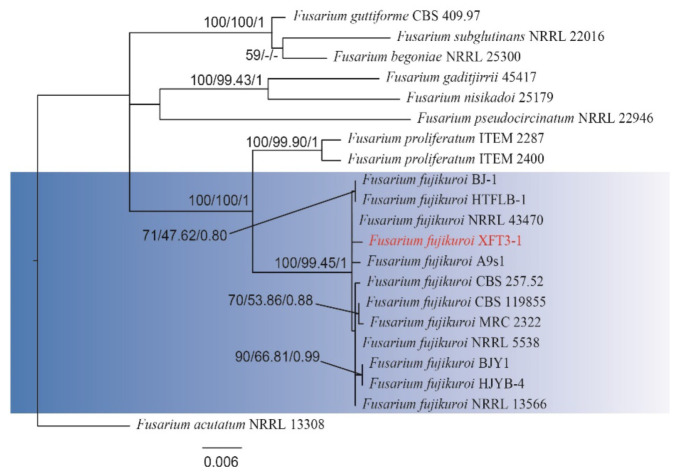
Maximum likelihood (ML) tree inferred from the combined sequence analysis of *RPB2*, *EF-1α*, and *TUB2* genes for selected *Fusarium* spp. Bootstrap support values and maximum parsimony (MP) and Bayesian **i**nference (BI) posterior probabilities are given at each node based on 1000 bootstrap replicates. The tree was rooted to *F. acutatum* NRRL 13308. The red text represents the causative agent of kiwifruit leaf spot caused by *Fusarium fujikuroi XFT3-1* in this study.

**Figure 3 jof-08-00239-f003:**
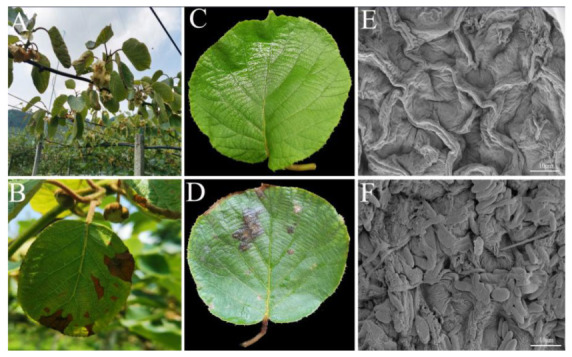
Natural field symptoms of kiwifruit leaf spot disease (**A**,**B**) symptoms 5 d after artificial inoculation ((**C**) inoculated with sterile water; (**D**) inoculated with 500 μL of 1 × 10^6^ conidia/mL), and microstructure (scanning electron microscope) of the upper epidermis of the leaves ((**E**) inoculated with sterile water; (**F**) inoculated with 500 μL of 1 × 10^6^ conidia/mL).

**Table 1 jof-08-00239-t001:** PCR primers for *EF-1**α*, *RPB2*, and *TUB2* gene amplification.

Target Sequence	Primer	Primer Sequence (5’-3’)	Reference
TEF	EF1	ATGGGTAAGGAGGACAAGAC	O’Donnell [26]
	EF2	GGAGGTACCAGTGATCATGTT	
RPB2	RPB2-5f2	GCCGTCAACGACCCCTTCATT	O’Donnell [27]
	RPB2-7cr	GGGTGGAGTCGTACTTGAGCATGT	
TUB2	Bt2a	GGTAACCAAATCGGTGCTGCTTTC	Glass and Donaldson [28]
	Bt2b	ACCCTCAGTGTAGTGACCCTTGGC	

**Table 2 jof-08-00239-t002:** Reference isolates used in this study and their GenBank accession numbers.

Species Name	Culture Collection Accession Numbers	Host/Isolate Source	GenBank Accession Number		
			TEF	RPB2	TUB
*Fusarium fujikuroi*	XFT3-1	Kiwifruit	OL774567	OL774568	OL774569
*Fusarium fujikuroi*	HJYB-4	Zanthoxylum armatum	MT902140.1	MT902141.1	MT902139.1
*Fusarium fujikuroi*	NRRL 13566	Yellow-eyed grass (*Xyris* spp.)	-	JX171570.1	U34415.1
*Fusarium fujikuroi*	HTFLB-1	Juglans sigillata	MN853324.1	MT909551.1	MT786729.1
*Fusarium fujikuroi*	BJ-1	Bletilla striata	MH263736.1	-	MH263737.1
*Fusarium fujikuroi*	NRRL 5538	Yellow-eyed grass (*Xyris* spp.)	MN193860.1	MN193888.1	-
*Fusarium fujikuroi*	A9s1	Soybean	MK560310.1	MN892319.1	-
*Fusarium fujikuroi*	MRC 2322	-	MH582343.1	MH582149.1	-
*Fusarium fujikuroi*	NRRL 43470	Fusarium keratitis	DQ790494.1	DQ790582.1	-
*Fusarium fujikuroi*	BJY1	Canna indica	-	MF984421.1	MF984415.1
*Fusarium fujikuroi*	CBS 257.52	Oryza sativa seedling	KU711678.1	KU604257.1	KU603885.1
*Fusarium fujikuroi*	CBS 119855	Environmental	MW401994.1	MW402735.1	MW402194.1
*Fusarium proliferatum*	ITEM2287		LT841245	LT841252	LT841243
*Fusarium proliferatum*	ITEM2400		LT841259.1	LT841266.1	LT841257.1
*Fusarium nisikadoi*	25179	Yellow-eyed grass (*Xyris* spp.)	MN193879.1	MN193907.1	-
*Fusarium gaditjirrii*	45417	Yellow-eyed grass (*Xyris* spp.)	MN193881.1	MN193909.1	-
*Fusarium pseudocircinatum*	NRRL 22946	Neotropical trees	MG838023.1	MN724939.1	MG838096.1
*Fusarium subglutinans*	NRRL 22016	-	HM057336.1	JX171599.1	-
*Fusarium begoniae*	NRRL 25300	-	MN193858.1	MN193886.1	-
*Fusarium guttiforme*	CBS 409.97	Population Genomic	MT010999.1	MT010967.1	MT011048.1
*Fusarium acutatum*	NRRL 13308	Yellow-eyed grass (*Xyris* spp.)	MN193855.1	MN193883.1	-

**Table 3 jof-08-00239-t003:** Inhibitory effect of different plant extracts on strain XFT3-1.

Natural Antifungal Agents	Concentrations (μg/mL)	Regression Equation	EC_50_ (mg/L)	r	95% Confidence Intervals
Osthole	25, 50, 100, 200, 400	Y = 1.3206 x + 2.6072	64.86 ± 0.18	0.9931	1.7497–56.1921
Cinnamaldehyde	20, 30, 40, 50, 60	Y = 2.23 x + 0.9631	64.60 ± 0.23	0.9936	1.7725–59.2310
Resveratrol	12.5, 25, 50, 100, 200	Y = 1.6667 x + 1.4861	128.33 ± 0.20	0.9928	2.0279–106.6450
Allicin	25, 50, 100, 200, 400	Y = 1.6466 x + 1.6437	109.22 ± 0.096	0.9949	1.9892–97.5529
Honokiol	20, 80, 150, 300, 450	y = 0.408 x + 4.4829	18.50 ± 0.20	0.9281	0.8312–6.7790
Citral	25, 50, 100, 200, 400	Y = 1.4632 x + 1.0391	509.25 ± 0.50	0.9601	2.4350–272.2776
Carvacrol	10, 30, 600, 100, 150	Y = 1.1128 x + 2.0262	470.21 ± 0.26	0.9743	2.3925–246.8687

## Data Availability

The datasets generated and/or analysed during the study are available from the corresponding author upon reasonable request.

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
