# Peer review of "Sensitivity Testing of Natural Antifungal Agents on Fusarium fujikuroi to Investigate the Potential for Sustainable Control of Kiwifruit Leaf Spot Disease"

_jof, 2022, doi:10.3390/jof8030239_

Round 1
Reviewer 1 Report
Abstract, too basic, no info about the problem,
you write that you found disease, and decided to identify, that's good, but is it economically significant disease??? what are yield losses?
line 33: kiwifruit1 change to kiwifruit
would be interesting to know how many (amount in tons or percentage) kiwifruit are grown in world and china. this would be good in introduction
line 35: not understand why there are numbers near diseases Latin names
39: what amount of yield losses?
line 40: incorrect Latin name F. fujikoroi
44 line: what means numbers near diseases name? its citing?
for which plants F. fujikoroi is important?
50-52 line: Based on the morpho-50 logical characteristics and the DNA sequences of three phylogenetic markers, this study 51 clarified the pathogenic factors of "Guichang" kiwifruit leaf spot disease and screened out 52 biological fungicides with fungistatic effects. its not an introduction it methods
the introduction must be improved
59 line: in methods, you write only Xifeng County samples, but in the introduction, much more Xifeng County, Guiyang City, Guizhou Province, China. Please add GPS coordinates, and all information, cultivar, BBCH
62 : PDA provider
66: you do not describe it the isolate's monocultures and if they were purified.
67-69 - from which plants, and what methods, no references
71: what temperature isolates were incubated, what conditions?
78-79: : making use of the translation elongation factor 1 alpha gene (EF-1É‘), RNA pol-78 polymerase II second largest subunit (RPB2) gene and beta-tubulin (TUB2) gene joint analysis. REferences????
80: thermocycler? References of PCR conditions?
93-94: not clear: you grow at various temperatures for how many days? and after then incubate ar 28 C 5 days and only then evaluate?????
how many replicates per treatment? how many isolates?
72: according to what you observed through a microscope? Reference?
93: PDA medium plate change to PDA plate. PDA is already medium
95: how diameter of the colony was measured?
97 : provider CZAPEKS
99: please clarify? add table or more details of with equal amounts of different carbon and nitrogen sources,
why you chose 28 C temperature?
107: how many strains were isolated? how many strains were in the experiments?
110: what amount was sprayed evenly on sterilized, healthy kiwifruit leaves.?
110: cultivar, from were got the heathy kiwifruit leaves.
111: leaves with petioles? or only leaves? in previous lines its written leaves
106: where experiment conducted? Petri? in vivo? boxes?
113: where leaves were stored? What temperature? Light? how inoculate? spayed in leave? injected?
119-121: from were extracted extracts? what concentrations were used?
124: what alternating light and dark conditions? PPFD? photoperiod? intensity light?
125-126: reference
133: chains 24. what 24? 24 mm ? 24 cm?
130: strain XFT3-1 is it fusarium??? no info in methods
how many days grew XFT3-1 strain measurements provided
139: what in pictures are small letters? Mi, Ma, Co
152: it should be in methods menthioned, and reference added
176: what letters EFe, Ee and ... indicate in the figure? no info in methods about carbon - nitrogen sources and concentrations
190: fig 4 A-B picture are from isolates? or you inoculated in field? Fig 4 C-D were the leaves were incubated? and A - C I assume are control, and B-D inoculated???? D-F what it shows infection??? add more description
192: no info about used fungicides in methods, add treatments
not clear the aim of this research
209-210: how many strains were tested in you experiments?>
206: once more, what numbers indicated at Lattin names
215-225: more induction that discussion
add discussion, its few sentences only
230: please add conclusions, how its like discussion
255: references not according to instructions
Author Response
I am very grateful to the reviewers for their suggestions. I have carefully considered your suggestions and made the following introductions and responses. If there are still questions, I will reply as soon as possible.
Comments and Suggestions for Authors: Abstract, too basic, no info about the problem
Thank you for your comments and suggestions. The Abstract section has reorganized the language.
Abstract: Kiwifruit is a nutritious and economically important fruit that is widely cultivated in China. In 2021, leaf spot disease of kiwifruit was discovered in the main kiwifruit-producing area of Xifeng County, Guizhou Province, China. Leaf spot disease weakens plant photosynthesis and reduces nutrient synthesis, thereby affecting plant growth. We studied the morphological characteristics and performed a combined analysis of EF-1É‘, RPB2, and TUB2 genes of Fusarium fujikuroi, a fungus associated with leaf spot disease. The pathogenicity of F. fujikuroi followed Koch’s hypothesis, confirming that this fungus is the cause of kiwifruit leaf spot disease. The sensitivity of seven natural antifungal agents against F. fujikuroi was measured using the mycelial growth rate method. Honokiol, cinnamaldehyde, and osthol showed good antifungal effects against F. fujikuroi, with EC50 values of 18.50, 64.60, and 64.86 μg/mL, respectively. The regression coefficient of cinnamaldehyde was the largest at 2.23, while that of honokiol was the smallest at 0.408. Fusarium fujikuroi was the most sensitive to cinnamaldehyde.
you write that you found disease, and decided to identify, that's good, but is it economically significant disease???
Thank for your question. Kiwifruit leaf spot is one of the important economically diseases in orchard. It mainly damages the leaves, necrosis, wilting and eventually leads to poor photosynthesis and hinders the synthesis of nutrients, resulting in a decrease in the yield and quality of kiwifruit. in May 2021, a kiwifruit with leaf spot disease was found in a kiwifruit plantation in Xifeng County, Guiyang City, Guizhou Province, China, with an incidence rate of approximately 23%.
line 33: kiwifruit1 change to kiwifruit
I have changed
would be interesting to know how many (amount in tons or percentage) kiwifruit are grown in world and china. this would be good in introduction
Thanks for your opinion. I have re-added the relevant kiwifruit introduction in introduction part
line 35: not understand why there are numbers near diseases Latin names
This article adopts the bibliographic format in numerical order, all numbers represent relevant references. its citing
39: what amount of yield losses?
The incidence of kiwifruit leaf spot is about 23% in field investigations, but, no statistics on actual yield losses
line 40: incorrect Latin name F. fujikoroi
Changed F. fujikoro to F. fujikuroi
44 line: what means numbers near diseases name? its citing? for which plants F. fujikoroi is important?
Thanks for your valuable comments. Numbers near diseases name is citing. F. fujikuroi is wide geographic distribution and host range, it is one of the most dif-ficult agricultural pathogens to control, and the relevant introduction is added again in the introduction of the article.
50-52 line: Based on the morpho-50 logical characteristics and the DNA sequences of three phylogenetic markers, this study 51 clarified the pathogenic factors of "Guichang" kiwifruit leaf spot disease and screened out 52 biological fungicides with fungistatic effects. its not an introduction it methods. the introduction must be improved
thanks for your valuable advice, I have removed lines 50-52. change to: introduction.
59 line: in methods, you write only Xifeng County samples, but in the introduction, much more Xifeng County, Guiyang City, Guizhou Province, China. Please add GPS coordinates, and all information, cultivar, BBCH
Thank you for your valuable suggestions. I have corrected the following information: in May 2021 (kiwifruit flowering period determined using the BBCH-scale), leaves of Actinidia deliciosa ‘Guichang’ visibly affected by leaf spot disease were collected from Xifeng County (27°2′N, 106°30′E).
62 : PDA provider
The PDA medium is prepared by itself, and the main components are: 200.0 g potato, 20.0 g glucose, 17.0 g agar/L
66: you do not describe it the isolate's monocultures and if they were purified.
Thanks for your opinion. I re-added the description of the relevant steps. The added information is: a single colony was selected and transferred to new PDA plates until the mycelia were identical in colour and shape
67-69 - from which plants, and what methods, no references
The natural antifungal agents (osthole, cinnamaldehyde, resveratrol, allicin, honokiol, citral, and carvacrol) used with purities of ≥98% were provided by Shanghai Macklin Biochemical Co., Ltd. (Shanghai, China) and stored at 4 °C.
71: what temperature isolates were incubated, what conditions?
Thanks for your input, I have added the information as: after incubation at 28 ℃ for 5 d in the dark.
78-79: making use of the translation elongation factor 1 alpha gene (EF-1É‘), RNA pol-78 polymerase II second largest subunit (RPB2) gene and beta-tubulin (TUB2) gene joint analysis. REferences????
There are relevant references in the text, which have been reflected in the text and in the table. O’Donnell et al [26], O’Donnell et al [27] and Glass and Donaldson [28].
80: thermocycler? References of PCR conditions?
There are relevant references in the text, O’Donnell et al [26], O’Donnell et al [27] and Glass and Donaldson [28]
72: according to what you observed through a microscope? Reference?
The references of microscope observation are mainly reflected in the result part.
Morphological identification of strain XFT3-1 as F. fujikuroi is consistent with the re-sults of Laurence [33] and Ibrahim [34].
93: PDA medium plate change to PDA plate. PDA is already medium
Thanks for the edit. medium has been removed
93-94: not clear: you grow at various temperatures for how many days? and after then incubate ar 28 C 5 days and only then evaluate?????how many replicates per treatment? how many isolates?
As suggested by another reviewer, this content has no significant effect on this article. I take this suggestion and delete this part.
95: how diameter of the colony was measured?
This modification is the same. As suggested by another reviewer, this content has no significant effect on this article. I take this suggestion and delete this part.
97 : provider CZAPEKS
This modification is the same. As suggested by another reviewer, this content has no significant necessityon this article. I take this suggestion and delete this part.
99: please clarify? add table or more details of with equal amounts of different carbon and nitrogen sources, why you chose 28 C temperature?
This modification is the same. As suggested by another reviewer, this content has no significant necessity on this article. I take this suggestion and delete this part.
107: how many strains were isolated? how many strains were in the experiments?
A total of 25 strains fungus with different colony morphology were isolated and used for Koch’s hypothesis.
110: what amount was sprayed evenly on sterilized, healthy kiwifruit leaves.? cultivar, from were got the heathy kiwifruit leaves.
Take 500 μL, 1 × 106 conidia/mL with a hemacytometer and spray evenly on the steri-lized healthy "Guichang" kiwifruit leaves with petioles. While 500 μL of the sterilised distilled water was inoculated as the blank control.
111: leaves with petioles? or only leaves? in previous lines its written leaves
Modified to leaves with petioles
106: where experiment conducted? Petri? in vivo? boxes?
The inoculated leaves were placed in a light incubator at 28 °C and 75% relative humidity with a 16/8 h light/dark photoperiod, and the disease progression of the leaves was regularly observed
113: where leaves were stored? What temperature? Light? how inoculate? spayed in leave? injected?
500 μL of 1 × 106 conidia/mL was collected using a hemacytometer and sprayed evenly on sterilised healthy Guichang kiwifruit leaves with petioles. For a blank control, 500 μL of sterilised distilled water was sprayed. Each petiole was wrapped in wet cotton to prevent the leaves from drying out. The inoculated leaves were placed in a light incubator at 28 °C and 75% relative humidity with a 16/8 h light/dark photoperiod, and the disease progression of the leaves was regularly observed.
119-121: from were extracted extracts? what concentrations were used?
Natural extracts were purchased from Shanghai Macklin Biochemical Co., Ltd. (Shanghai, China). The specific concentrations are listed in Table 3
124: what alternating light and dark conditions? PPFD? photoperiod? intensity light?
cultured for temperature of 28 ℃ and 75% relative humidity for five days under dark conditions.
133: chains 24. what 24? 24 mm ? 24 cm?
“[24]” is citing
130: strain XFT3-1 is it fusarium??? no info in methods. how many days grew XFT3-1 strain measurements provided
Thanks for your advice. Description has been re-added. strain XFT3-1 is it F. fujikuroi .for 5 days
139: what in pictures are small letters? Mi, Ma, Co
A note below the title. Note: Ma: macroconidia, Mi: microconidia, Co: Conidiophore
152: it should be in methods menthioned, and reference added
References added
176: what letters EFe, Ee and ... indicate in the figure? no info in methods about carbon - nitrogen sources and concentrations
This modification is the same. As suggested by another reviewer, this content has no significant necessity on this article. I take this suggestion and delete this part.
190: fig 4 A-B picture are from isolates? or you inoculated in field? Fig 4 C-D were the leaves were incubated? and A - C I assume are control, and B-D inoculated???? D-F what it shows infection??? add more description
Thanks for your questions and suggestions. Make the following modifications:
Figure 3. Natural field symptoms of kiwifruit leaf spot disease (A, B), symptoms 5 d after artificial inoculation (C, inoculated with sterile water; D, inoculated with 500 μL of 1 × 106 conidia/mL), and microstructure (scanning electron microscope) of the upper epidermis of the leaves (E, inoculated with sterile water; F, inoculated with 500 μL of 1 × 106 conidia/mL).
192: no info about used fungicides in methods, add treatments . not clear the aim of this research
Information on antibacterial agents has been referred to in 2.1 Materials and methods for relevant source information
209-210: how many strains were tested in you experiments?
A total of 25 strains fungus with different colony morphology were isolated and used for Koch’s hypothesis.
206: once more, what numbers indicated at Lattin names
Numbers indicate corresponding references
215-225: more induction that discussion. add discussion, its few sentences only
Thanks for your opinion, I have reorganized the discussion
230: please add conclusions, how its like discussion
Thanks for your opinion, I have reorganized the conclusions
255: references not according to instructions
Thanks for your edits. References have been revised as required by the journal

Reviewer 2 Report
The paper presents interesting results on the effectiveness of different known eco-friendly substances to control plant diseases. The authors properly identify local Fusarium strains using molecular techniques and apply the before mention substances to select the most effective one. This makes the overall work interesting but there are several concerns that should be considered to increase the readability of the work.
- First, in line 86 it is mentions that “the tested gene sequences, which were uploaded to the GenBank database in NCBI”. Please, provide the GeneBank accession number for the sequences used for the taxonomy of the isolates.
- Second. Please, explain why different temperatures were tested (line 91 and figure 3). If was observed that a temperature of 30 was optimal (line 162) but nevertheless, a temperature of 28 was used for regular incubation. The data was ignored by the authors and it does not help for identification of the fusarium strains. The data seems superfluous, I would recommend to remove it.
- Third Please explain why nutritional conditions were evaluated. Similar as before, optimal nutritional medium conditions were ignored and the regular incubation of the fungus was done in PDA, SNA (line 71, for morphological analysis) or PDB (line 107, for inoculation assays). Similar as before, the data was ignored by the authors and it does not help for identification of the fusarium strains. The data seems superfluous, I would recommend to remove it.
- Four. In figure 4 central images should not be mention first (line 180, Fig 4C,D), if these images need to be mentioned first, they should be labeled as A and B.
- Five. In figure 4 what is the difference between fig 4C and 4D? is the leaf in fig 4C inoculated? Please clarify this in the text. The same for fig4 A and B. Is figure 4 A natural inoculation and B artificial inoculation? Please clarify this in the text
Author Response
I am very grateful to the reviewers for their suggestions. I have carefully considered your suggestions and made the following introductions and responses. If there are still questions, I will reply as soon as possible.
First, in line 86 it is mentions that “the tested gene sequences, which were uploaded to the GenBank database in NCBI”. Please, provide the GeneBank accession number for the sequences used for the taxonomy of the isolates.
Thanks for your comments, strain XFT3-1 accession numbers are in Table 2
Second. Please, explain why different temperatures were tested (line 91 and figure 3). If was observed that a temperature of 30 was optimal (line 162) but nevertheless, a temperature of 28 was used for regular incubation. The data was ignored by the authors and it does not help for identification of the fusarium strains. The data seems superfluous, I would recommend to remove it.
Thank you for your valuable suggestion, and I accept it, remove this paragraph
Third Please explain why nutritional conditions were evaluated. Similar as before, optimal nutritional medium conditions were ignored and the regular incubation of the fungus was done in PDA, SNA (line 71, for morphological analysis) or PDB (line 107, for inoculation assays). Similar as before, the data was ignored by the authors and it does not help for identification of the fusarium strains. The data seems superfluous, I would recommend to remove it.
Thank you for your valuable suggestion, and I accept it, remove this paragraph
Four. In figure 4 central images should not be mention first (line 180, Fig 4C,D), if these images need to be mentioned first, they should be labeled as A and B.
Thanks for your opinion. I have modified the corresponding labeled.
Five. In figure 4 what is the difference between fig 4C and 4D? is the leaf in fig 4C inoculated? Please clarify this in the text. The same for fig4 A and B. Is figure 4 A natural inoculation and B artificial inoculation? Please clarify this in the text
Thanks for your suggestion, I have re-described in the text. Figure 3. Natural field symptoms of kiwifruit leaf spot disease (A, B), symptoms 5 d after artificial inoculation (C, inoculated with sterile water; D, inoculated with 500 μL of 1 × 106 conidia/mL), and microstructure (scanning electron microscope) of the upper epidermis of the leaves (E, inoculated with sterile water; F, inoculated with 500 μL of 1 × 106 conidia/mL).

Round 2
Reviewer 1 Report
please revise the article there are some mistakes with references, I see in some places: Error! Reference source not found.